# Research Status and Prospect of Non-Viral Vectors Based on siRNA: A Review

**DOI:** 10.3390/ijms24043375

**Published:** 2023-02-08

**Authors:** Liangnan Tong, Danqing Liu, Zhiyue Cao, Nannan Zheng, Chenchen Mao, Shujuan Liu, Liangcan He, Shaoqin Liu

**Affiliations:** 1School of Material Science and Chemical Engineering, Harbin University of Science and Technology, Harbin 150040, China; 2School of Life Science and Technology, Harbin Institute of Technology, Harbin 150001, China; 3School of Medicine and Health, Harbin Institute of Technology, Harbin 150001, China; 4Department of Electrical, Computer and Energy Engineering, University of Colorado Boulder, Boulder, CO 80309, USA; 5School of Materials Science and Engineering, Harbin Institute of Technology, Harbin 150001, China; 6Key Laboratory of Microsystems and Microstructures Manufacturing (Ministry of Education), Harbin Institute of Technology, Harbin 150001, China

**Keywords:** gene therapy, non-viral vectors, siRNA delivery, biomedical application

## Abstract

Gene therapy has attracted much attention because of its unique mechanism of action, non-toxicity, and good tolerance, which can kill cancer cells without damaging healthy tissues. siRNA-based gene therapy can downregulate, enhance, or correct gene expression by introducing some nucleic acid into patient tissues. Routine treatment of hemophilia requires frequent intravenous injections of missing clotting protein. The high cost of combined therapy causes most patients to lack the best treatment resources. siRNA therapy has the potential of lasting treatment and even curing diseases. Compared with traditional surgery and chemotherapy, siRNA has fewer side effects and less damage to normal cells. The available therapies for degenerative diseases can only alleviate the symptoms of patients, while siRNA therapy drugs can upregulate gene expression, modify epigenetic changes, and stop the disease. In addition, siRNA also plays an important role in cardiovascular diseases, gastrointestinal diseases, and hepatitis B. However, free siRNA is easily degraded by nuclease and has a short half-life in the blood. Research has found that siRNA can be delivered to specific cells through appropriate vector selection and design to improve the therapeutic effect. The application of viral vectors is limited because of their high immunogenicity and low capacity, while non-viral vectors are widely used because of their low immunogenicity, low production cost, and high safety. This paper reviews the common non-viral vectors in recent years and introduces their advantages and disadvantages, as well as the latest application examples.

## 1. Introduction

Traditional methods of treating diseases include drug treatment. The bioavailability of the drug is reduced because the drug has experienced metabolic disorders and incomplete drug absorption [1]. In addition, the reverse effect and sudden release of drugs on healthy cells also lead to various side effects. Gene therapy is a promising therapeutic strategy [2], which can be operated at the gene level to address the occurrence of diseases fundamentally [3] and provide more possibilities and flexibility than drug therapy [4]. Gene therapy will make great achievements in areas where there is no treatment or poor efficacy at present [5], especially in neurodegenerative diseases [6,7], congenital genetic diseases [8,9], and malignant tumors [10,11]. Among the reported methods for gene delivery, RNA interference (RNAi) has been studied and developed extensively [6]. RNAi refers to the phenomenon that small double-stranded RNA can specifically degrade or inhibit the expression of homologous mRNA, thus inhibiting or shutting down the expression of specific genes [12,13,14,15]. Double-stranded RNA is cleaved by enzymes to form many small fragments called siRNA. By revealing the pathogenic gene of a particular disease, siRNA can be designed to target the mRNA of the gene and lyse it to achieve the purpose of curing the disease [16,17]. Molecular therapy by siRNA has attracted widespread attention because of its great potential in the treatment of various cancers, viral infections, genetic diseases, and pain management [18]. Undoubtedly, it is one of the most popular research fields in life science at present [19] and one of the most promising areas for drug development in the future.

In 1998, Mello et al. injected the mRNA-encoding muscle proteins into *C. elegans*, where the mRNA carrying the genetic code is called the “justice mRNA,” and the mRNA carrying the antigenetic code information is called the “antisense mRNA.” They found that the double-stranded RNA (dsRNA), originally added as a control, had significantly higher expression of the suppressor gene than the other groups [20]. They demonstrated that dsRNA could effectively reduce the expression activity of the target genes. When dsRNA enters the cell, it is cleaved by a ribozyme and degraded into small RNA fragments with a length of 21–23 nt, called siRNA. Subsequent siRNA binds to an RNA-induced silencing complex (RISC) in the cytoplasm and unbinds to a single strand. As the justice strand is degraded, the remaining antisense strand guides RISC to bind to the complementary mRNA, causing RISC to cut off and degrade the mRNA. It laid the foundation for the development of siRNA [21].

However, intravenous free siRNA cannot readily be absorbed into cells and can be degraded by nucleases in plasma or tissues at any time [22]. Renal clearance and uptake by phagocytes will lead to a lower content of siRNA in vivo, and free siRNA cannot penetrate the lipid membrane of cells, so the therapeutic effect is not significant [23]. In recent years, it has been demonstrated that using nanoparticles as gene delivery carriers may effectively silence genes while protecting siRNA and having high biocompatibility and biodegradability [24]. BNT162b2 vaccine is an RNA vaccine targeting the coronavirus in 2019. BNT162b2 is a lipid nanoparticle–formulated, nucleoside-modified RNA vaccine that encodes a prefusion stabilized, membrane-anchored SARS-CoV-2 full-length spike protein. The study shows that the two-dose injection scheme of BNT162b2 has a 95% protective effect against COVID-19 in people aged 16 years or over. Safety over a median of 2 months was similar to that of other viral vaccines [25,26,27]. Clinical delivery systems currently in use can be divided into two categories: viral and non-viral vectors. Although viral vectors have high gene transfection efficiency, they still face inevitable limitations and risks. Compared with non-viral vectors, the production process of viral vectors is complex, and the production cost is high, which is not suitable for mass production [28,29]. The potential mutagenicity and adverse immune characteristics of viral vectors limit their application in gene therapy and inhibit the potential of gene therapy [30,31]. People have been committed to developing non-viral vectors with better security and effectiveness. Carriers based on liposomes [32,33], inorganic [34], and polymer nanoparticles have been considered and studied for delivering siRNA [35,36]. At present, three siRNA drugs have been approved by FDA for marketing, including Paisiran (trade name: Onpattro), Givosiran (trade name: Givlaari), and Lumasiran (trade name: Oxlumo); the names, company information, and treatment diseases of siRNA drugs that have been approved for marketing and are undergoing phase III clinical trials are shown in Table 1 [37].

The siRNA therapeutics also present significant challenges, including safety and stability in vivo as well as the delivery of siRNA to the required cells and organs [38,39,40], which makes nanomaterial design more challenging. Therefore, the design of the vector should not only minimize harm to the human body but also facilitate the delivery of siRNA while overcoming the delivery challenges [41]. Figure 1 shows the transmission process of non-viral vector-loaded siRNA in vivo. The delivery of siRNA nanocarriers to specific therapeutic cells can be accomplished in two ways: passive targeting and active targeting. The enhanced permeability and retention (EPR) effect of solid tumors refers to the tendency of molecules or particles of certain sizes to gather in tumor tissues as compared to normal tissues [42,43]. The reason for this is that the microvascular endothelial gaps have large densities and complete structures in normal tissues. As a result, macromolecular and lipid particles cannot easily pass through the blood vessel walls. Furthermore, in solid tumor tissues, there are many blood vessels, a wide gap in the hemal wall, poor structural integrity, and a lack of lymphatic circumfluence; therefore, the macromolecular material and lipid particles are highly selectively permeabilized and retained [44,45,46,47,48,49,50]. Most tumor cells exhibit the EPR effect, making it a crucial breakthrough for nano-drugs targeting tumors [51]. Additionally, siRNA’s active targeting depends on the nanocarrier’s modification [52]. In contrast to normal cells, tumors’ surfaces have receptors or targets that can be targeted by nanocarriers vis drug delivery through the interaction between specific receptors and ligands, thus improving the therapeutic effect of drugs [53].

Once it reaches the targeted tissue, the most critical three processes are penetrating the cell membrane [54], escaping from the endosome or lysosome [55], and releasing siRNA into the cytoplasm [56]. Typically, the macromolecules and granular substances enter cells mainly through endocytosis [57] and membrane fusion [58]. Once these substances attach to the cell membrane, they are enclosed within the vesicles as the membrane invaginates to form vesicles. After that, vesicles are separated from the cell membrane and enter the cell interior [59,60,61,62]. If it fails to get out of the endosome in time, it gets broken down inside the cell. Therefore, endosomal or lysosomal escape is also critical to siRNA-mediated gene silencing [63]. Some carriers, such as cationic liposome trimethyl-2 bromide, and 3-dioleoxypropyl ammonium (DOTAP), enter the endosome through endocytosis. It is known that the accumulated ion pairs can destroy the stability of the endosomal membrane and cationic lipid-coated carriers [64] after the formation of ion pairs mediated between the positively charged group of DOTAP and the negatively charged group of the endosomal membrane. Moreover, some cationic nanocarriers can facilitate the escape of siRNA from endosomes or lysosomes through the proton sponge effect [65]. The unsaturated amino chelate on the particles is provided by the proton pump, which keeps the proton pump open, and each proton causes chloride ions and water molecules to remain in the lysosome, which eventually causes the lysosome to swell and break, releasing the particles into the cytoplasm [66,67,68,69,70,71,72]. The specific concentration of ATP in the cytoplasm contributes to the release of siRNA.

As an emerging gene therapy tool, siRNA has gained wide research interest due to its high gene silencing efficiency and low toxicity. However, the clinical application of siRNA faces critical challenges, such as low delivery efficiency of siRNA, susceptibility to degradation, and inability to target tumor tissues. In this context, carriers that can efficiently deliver siRNA are needed. The high immunogenicity of viral vectors and the high production cost limit their application. Delivery techniques based on liposomes, inorganic, and polymeric nanoparticles in nonviral vectors are gradually being developed. This review introduces the development of non-viral vector systems used for siRNA delivery, focuses on the advantages and disadvantages of different non-viral vector systems, summarizes the results and effects of the latest application of different non-viral vector systems loaded with siRNA in a wide range of diseases, and discusses its research prospects.

## 2. siRNA Nanocarriers

To maximize the effect of gene silencing [73,74], siRNA vectors should be selected discreetly to ensure that they can integrate siRNA and combine with other substances. In this section, we mainly introduce several common carriers, such as liposomes, inorganic, and polymeric nanoparticle material platforms.

### 2.1. Liposomes

Liposomes are a new therapeutic formulation capable of targeted drug delivery [75,76,77]. One of the main chemical components of liposomes is a phospholipid, among which the most representative phospholipid is lecithin. It is a kind of neutral phospholipid, and it has the advantages of low preparation cost and high stability. Liposomes are primarily made up of phosphatidylcholine, which can also be found in liposomes as the main raw material [78]. Cholesterol (CHO) is another key component of liposomes. As the main component of many natural biofilms, CHO does not form the membrane structure itself but can be incorporated to stabilize the phospholipid bilayer membrane because it can change the permeability and fluidity of the membrane when added [79]. As the siRNA is transported through the body, the lipid bilayer protects it from being broken down [80]. Due to the similarity of membrane structures between liposomes and cells, liposomes are also highly biocompatible and biodegradable [81] and can store a variety of substances, making them suitable carriers for drug delivery in vivo. In the following section, two commonly used liposomes, traditional liposomes and cationic liposomes, will be briefly introduced.

#### 2.1.1. Conventional Liposome

The first US FDA-approved siRNA vector for siRNA therapy was liposome, thanks to its low toxicity and biodegradability. Conventional liposomes use phospholipids and cholesterol as membrane materials. It is prepared by traditional methods (such as injection, film dispersion, freeze-drying, reverse phase evaporation, and hydration methods) [82]. The liposome prepared by the above method comprises multilayer vesicles and monolayer vesicles. Multi-layer vesicles are like onions with a diameter of 100–800 nm. When the phase change temperature is above, ultrasound is applied to the multilayered vesicles to form a single-layer vesicle with a diameter of 30–60 nm and a wall thickness of about 5 nm. When the particle size of liposomes is less than 50 nm, they can generally target spleen tissue; it can target liver tissue at 50–100 nm; 0.1–0.2 μm can target lysosomes of liver macrophages in liver tissue; Between 7–12 μm can be absorbed by lung tissue cells. When the particle size is 12 μm can be absorbed by capillary epithelial cells and then reach the tumor-bearing tissues; when the particle size is greater than 15 μm, it can be absorbed by vascular epithelial cells such as mesenteric artery [83].

Patel et al. developed a lipophilic preparation of lopinavir. The average particle size is 659.7 ± 23.1 nm. The drug encapsulation efficiency of the liposome reached about 89%. The oral bioavailability of lopinavir was increased by 2.2 times by the precursor lipid preparation [84].

Ghassemi et al. prepared carvedilol liposomes enriched with surfactants by membrane hydration technology. The drug encapsulation rate reached 79.8%. The results showed that the oral bioavailability and therapeutic effect of hydrophobic drugs with extensive systemic pre-metabolism could be greatly improved by preparing liposomes rich in surfactants. It showed a 2.3-fold increase in bioavailability [85].

Kim et al. synthesized lyophilized Eudragit-coated liposomal docetaxel (DTX) by freeze-drying. The half-life of oral liposome preparation was longer (t_1/2_), and the oral bioavailability was increased by three times. Therefore, freeze-dried Eudragit-coated liposome DTX may be a promising method for the treatment of various solid tumors [86].

Liposomes can be used as antibodies, aptamers or other receptor ligands. The surface modification strategies and classification methods of liposomes are shown in Figure 2 [87]. Conventional liposomes can be used to treat diseases that are associated with mono-macrophage system cells. Despite the wide use of phospholipids, their derivatives, e.g., dioleylphosphatidyl ethanolamine (DOPE), can promote the release of siRNA [88]. Moreover, CHO derivatives, such as metform in CHO derivatives, utilize the fusion activity of the part of the Formin and its inherent ability to induce tumor apoptosis to achieve the combined anti-carcinogenic effect [89]. However, a major problem with this kind of strategy is that siRNA-containing liposomes are easily degraded by lysosomal enzymes, including phospholipases after endocytosis, where liposomes bind to lysosomes. This means that siRNA release must be achieved before the liposomes are destroyed by lysosomes [90].

For this, Fuita et al. investigated the effect of curcumin on the content released by liposomes for the first time. In their work, DOPC/DOPE, DOPC, and DOPC/Cholesterol liposomes were used, and DOPC/DOPE liposomes were selected because they have been proven to be capable of carrying polynucleic acid. The properties of DOPC/DOPE, DOPC, and DOPC/Cholesterol liposomes are shown in Table 2. According to the report, DOPE will increase the permeability and fusion of the DOPC bilayer, while cholesterol will reduce these characteristics. The corresponding siRNA release capacity is DOPC/DOPE > DOPC > DOPC/Cholesterol liposomes. Curcumin promotes the release of siRNA from liposomes because it helps to regulate the permeability of liposomes and further release siRNA. In addition, the effect of temperature on curcumin promoting the release of siRNA from liposomes was further explored [91].

#### 2.1.2. Cationic Lipid (CL)

To maximize the loading rate of siRNA and improve the stability of blood circulation, positively charged cationic liposomes were developed [92,93]. It has been proven that cationic liposomes are the most promising siRNA carriers with high load rates, biocompatibility, and biodegradability. Commonly used cationic lipids include DOTAP [94], trimethyl-2 chloride, 3-dioleoxypropyl ammonium (DOTMA) [95], 3β-[N- (N′, N′-dimethylaminoethyl) aminoformyl] CHO (DC-CHOl) [96], etc. However, the stability, membrane fusion, and transfection efficiency of liposomes alone are poor, so auxiliary lipids are often necessary. The most commonly used adjuvant lipid is DOPE [97]. Its fusion and transfection efficiency are higher than dioleoylphosphatidylcholine (DOPC) and CHO liposomes [98]. Due to its negatively charged property, siRNA is absorbed by CL through electrostatic interaction [99]. Co-loading chemotherapy drugs and siRNA into cationic liposomes has shown significant advantages in cancer treatment. MCF-7 cells co-loaded with paclitaxel (PTX), crizotinib (CRI), and Bcl-XLsiRNA in cationic liposomes showed a significant EPR effect in breast cancer [100]. Using cationic liposomes to encapsulate antibiotics can reduce the half-life of antibiotics and reduce their side effects. The encapsulation rate of cefepime can be greatly improved by adding a cationic surfactant. Additionally, cefepime exhibits high antibacterial activity when loaded into cationic liposomes, especially against *E. coli*, which is also conducive to the release of antibiotics [101]. Despite the above-mentioned advantages, there is also concern over the clinical application of CL. They may cause toxicity and immunogenicity. These issues can be solved by the functionalization of cationic polymers.

### 2.2. Polymers

As an important non-viral carrier, cationic polymers have the characteristics of easy synthesis and flexibility [102,103]. More and more studies have shown the effectiveness of using polymers as siRNA vectors for gene therapy. They can be synthetically or chemically modified to improve gene transfection efficiency and reduce cytotoxicity [104]. Compared with neutral polymers, cationic polymers have better targeting capability [105,106]. Moreover, cationic polymers are positively charged, caused by the amine group in their structures. The electrostatic interaction between the positively charged cationic polymer and negatively charged siRNA led to the formation of polyelectrolyte complexes [107].

#### 2.2.1. Biopolymers

Biopolymers and synthetic polymers are currently used in gene therapy [108]. Biopolymers as gene therapy vectors have the advantages of good biocompatibility, biodegradability, and low toxicity. The most widely used natural biodegradable polymers are natural polysaccharides [109,110], nanocarriers based on peptides or proteins [111], and DNA [112].

Natural polysaccharides are biocompatible, biodegradable, and non-toxic, making them great candidates for drug delivery. Among them, chitosan (CS) and cyclodextrin (CDs) are commonly used [113,114,115] and have demonstrated excellent efficacy in siRNA-mediated gene silencing. Alternatively, polysaccharides can also be combined with other vectors to reduce toxicity.

The primary amine group of the main chain of CS is easily protonated under acidic conditions and combined with siRNA electrostatically [116], which plays an important role in tumor treatment [117]. In recent years, CS has been widely used in gene therapy [112]; however, unmodified CS, due to the low proton-buffering capacity and poor endosomal escape of this material, has limited its use in clinical practice [118]. The reduced charge of CS will cause it to lose stability when loaded with siRNA in blood. To solve this problem, CS derivatives, such as alkyl chain [119], PEI (Figure 3) [120], and CS modified by PEG [121], are proposed as solutions [122]. Mobarakeh et al. designed CS-PEG-carboxymethyl dextran to deliver siRNA, which was proven to effectively knock out part of HIV genes and prevent drug resistance [123].

As a natural polymer with a loose and porous structure and biocompatibility, CDs are another good option for siRNA delivery [124,125]. While it should be noted that natural CDs create unstable compounds with nucleic acids, as a result, they are not efficient agents by themselves for gene delivery. However, CDs have several unique features: (1) hydrophilic outer surfaces, (2) hydrophobic cavities, and (3) a significant number of hydroxyl groups on the ring surface that can be modified with different functional groups [126], forming cationic, amphiphilic, or pegylated CDs, which provides an effective way for nucleic acid transmission. Modified CDs are widely used to produce gene vectors with high loading rates and high targeting rates [127,128,129].

Evans et al. developed an amphiphilic vector targeting prostate cancer cells using DSPE-PEG5000-folic acid-modified CD. The vector can effectively protect siRNA from degradation. A component of the vector, GALA is a 30 amino acid endosomal escape peptide derived from viral proteins that enhance endosomal release following its uptake. The findings of this study demonstrated that the transcription factors and protein levels of ZEB1 and NRP1 were significantly reduced, which offers a broad range of potential applications in the treatment of prostate cancer [130].

Huntington’s disease is a dominant neurodegenerative disease. Like most neurological diseases, Huntington’s disease lacks specific treatment methods. There is currently no effective drug to delay the onset of Huntington’s disease symptoms. Mendonca et al. designed a novel delivery method, which is based on modified CD nanoparticles loaded with siRNA targeting huntingtin (HTT) gene and combined with rabies virus glycoprotein (RVG is a BBB shuttle peptide). The in vitro model showed that the CD platform could successfully penetrate brain endothelial cells and release siRNA into the cytoplasm of neuronal cells, further mediating the downregulation of HTT. The platform has a wide range of therapeutic potentials in other diseases, with gene validation targets in the central nervous system [131].

Alarcón et al. synthesized amphiphilic CDs (ModCBHD) and DNA-DOX complexes as supramolecular anticancer drug carriers. ModCBHD vesicles have a positively charged surface and can be used for encapsulating the negatively charged DNA-DOX. This system showed great stability over a long period, small particle size, excellent drug loading capability, and strong, controlled release ability. These facts could be exploited to take advantage of EPR effects and decrease the side effects of DOX [132].

During the process of targeted drug delivery of siRNA, biopolymer materials can protect siRNA from degradation, extending blood circulation time and achieving targeted drug delivery to tissues or organs. Biopolymers can also reduce immunogenicity while maintaining efficient siRNA and drug delivery. Biopolymers degradation products are not only non-toxic to the human body but also can be absorbed and metabolized by the human body.

#### 2.2.2. Synthetic Polymers

By utilizing various polymer groups, synthetic polymers have been developed as carriers with high transfection efficiency and minimal toxicity [133,134], great stability, high loading rate for siRNA, and small particle size, leading to their great potential in drug delivery. The vectors commonly used in gene therapy include polyethylene imine (PEI) [135], poly-lysine (PLL) [136], polylactic acid hydroxyacetic acid copolymer (PLGA) [137], and polyamide-amine dendritic polymers (PAMAM) [138], poly(amine-co-ester) (PACE), etc.

PEI was first developed in 1995. There are two forms of PEI: linear (lPEI) and branched (bPEI), which later became the most common cationic polymer due to its low cost, high transfection efficiency, and high intracellular body escape efficiency [139]. Due to the presence of amine groups in PEI, the extra amine matrix in the endosome will be ionized. When PEI is electrified, PEI can cause a “proton sponge effect” conducive to the escape of the endosome [140]. The molecular weight of PEI is positively correlated with its gene transfection efficiency, and PEI with high molecular weight shows high transfection efficiency [141]. In particular, PEI with a high molecular weight of 25 kDa is the “gold standard” for determining the transfection efficiency of non-viral vectors [142]. However, high cation density results in severe toxicity due to the inherent non-degradability [143]. Although low molecular weight PEI is less harmful, low molecular weight PEI produces poor gene transfection efficiency. To solve this problem, many approaches have been adopted to reduce the cytotoxicity of PEI in the practical application of gene delivery.

To construct amphiphilic PEI coupling, either biodegradable or synthetic materials were cut off into PEI, or hydrophobic fragments were introduced into PEI [144,145]. Ansari et al. proposed that the substitution of hydrophobic groups had a great influence on the properties of polymers. Low-molecular-weight PEI was modified with CHO and yielded water-insoluble polymers. Characterization results revealed that CHO-modified PEI had a significantly smaller size and an enhanced siRNA loading capacity compared with pure PEI. The good therapeutic effect of increased cell uptake was observed in the treatment of chronic myeloid leukemia [146]. By modifying PEI with tyrosine, Karimov et al. were able to significantly enhance the transfection efficiency of siRNA, reduce PEI-induced toxicity, improve biocompatibility, and also show positive therapeutic effects in a mice tumor model. It has also been demonstrated that fluorination is a successful means of lowering the toxicity of PEI [147]. Xue et al. introduced two series of fluorinated PEI via amidation with ethyl trifluoroacetate and perfluorobutyryl chloride, and Fluorination substantially improved the performance of PEI for siRNA delivery by reducing the cytotoxicity to MDA-MB-231 cells. (Figure 4) [143].

The FDA-approved biodegradable functional organic compound known as Polylactic acid-glycolic acid copolymer (PLGA) is synthesized by polymerizing lactic acid and glycolic acid [148]. It will be broken down into lactic acid and glycolic acid in the human body for absorption and metabolism, and finally, it will form carbon dioxide and water and be discharged out of the body. It has been widely used because of its non-toxicity, good transfection efficiency, and biocompatibility, which enable it to protect genetic material from degradation [149].

As PLGA is easily cleared by the immune system and lacks a specific targeting effect, it has limited its clinical application. PEG has been shown to protect PLGA from degradation, increasing drug permeability and thus enhancing gene transfection efficiency [150]. At the same time, chemotherapy drugs, siRNA, and PLGA can be used in combination to treat cancer [151]. Wu et al. reported that PLGA-cationic lipids nanocomposites deliver siRNA to treat severe lung diseases. In addition to preventing siRNA from degradation during delivery, PLGA also lessens toxicity caused by the positive charges of cationic lipids [152]. To improve the targeting and uptake rate of PLGA, Ghareghomi et al. designed folic acid functionalized PLGA and loaded Wortmannin and hTERT siRNA into PLGA-FANPs, loading MNPs for magnetic resonance imaging. A new strategy for the treatment of ovarian cancer has been made possible by studies demonstrating that the combination of drugs and siRNA can promote the apoptosis of cancer cells [153]. Miele et al. used PLGA as the core surface coated CS oleic acid as the carrier; the interaction between negatively charged oligonucleotide and positively charged amine on the surface of CS protected siRNA from degradation and contributed to endocytosis and endosomal escape [154].

PAMAM, which has a large number of primary and tertiary amine groups in its structure and a favorable proton sponge effect, is the first reported dendritic macromolecule [155,156]. PAMAM is considered to be an effective gene therapy vector due to its cationic properties and surface functional groups. Therefore, the use of PAMAM as a vector to treat cancer and overcome multidrug resistance of tumors has received increasing attention [157]. Compared with PEI, PAMAM has gradually gained popularity in various fields because of its notable biocompatibility and size flexibility [156]. PAMAM can be used as a siRNA carrier mainly because PAMAM protonation is easy to combine with negatively charged siRNA through electrostatic interaction to form a complex [158]. Secondly, the amines in the structure have a high proton buffering capacity that enables the polymers to escape from the lysosome [159].

PAMAM has high transfection efficiency. However, its toxicity, lack of targeting, and low capacity for siRNA release are all important factors that need a solution [160,161]. To address this issue, PAMAM is usually modified by introducing PEG, folic acid, and polypeptides [162]. Following the modification of the cyclic arginine-glycine-aspartic acid (cRGD) peptide, PAMAM exhibits lower cytotoxicity and is more concentrated in cancer cells [163].

Pan et al. used PAMAM and PEG-phospholipid polymers as carriers and treated multidrug resistance with DOX in ovarian and breast cancer cells. They showed that the nanocarriers reduced P-GP in MDR cancer cells and enhanced DOX [164]. Using a pH-sensitive LP shell and dendritic macromolecule PAMAM, Hu et al. designed a multifunctional liposomal nanocarrier that delivered DOX and PD-L1-targeting siRNA simultaneously. The results illustrated that siPD-L1@PM/DOX/LPs had suitable Zeta potential and particle size and showed stability in blood circulation. In vitro, experimental results demonstrated that DOX could be effectively released into the cytoplasm. According to in vivo findings, siPD-L1@PM/DOX/LPs had a high concentration at the tumor site and significantly inhibited tumor growth compared with other controls. The nano-carrier is expected to be used in the clinical treatment of tumors in the future [165].

Diabetic foot ulcers are a serious chronic complication in diabetic patients. The study showed that the high expression of MMP-9 in the wound damaged the balance between extracellular matrix (ECM) synthesis and degradation, resulting in delayed wound healing. However, the current MMP-9 inhibitors have shortcomings, such as poor specificity and high price. Luo et al. made a polyaminoamine-conjugated dextran conjugated with poly(amidoamine) (Dextran-PAMAM) to deliver siRNA targeting MMP-9 (siMMP9). The rat wound model showed that the compound could effectively inhibit the expression of MMP-9 and accelerate wound healing. It provides a new direction for developing new therapeutic dressings to promote diabetes wound healing [166].

PLL is far less hazardous than high molecular weight or high charge polymers, but it also has less ability to bind to siRNA [167]. Studies have shown that natural polyphenols enhance their ability to bind with nucleic acids. Fan et al. grafted synthesized cationic polycatechin onto PLL to increase siRNA loading. Subsequent tests showed that it successfully silenced macrophage tumor necrosis factor and achieved a positive therapeutic effect on ulcerative colitis without causing various adverse reactions (Figure 5) [168]. Yang et al. combined melanin with PLL to form stable polymers. PLL binds to siRNA through electrostatic interaction. Melanin acts as a photothermal sensitizer, inducing endosome escape and promoting siRNA to enter the cytoplasm. Polymers loaded with anti-survivin siRNA significantly slow down tumor growth both in vitro and in vivo [169].

Biodegradable poly (amine copolyester) (PACE) nanoparticles have great prospects in the transmission of siRNA. PACE was synthesized by enzymatic copolymerization of 15-pentanelactone (PDL), diethyl sebacate (DES), and N-methyldiethanolamine (MDEA). The advantage of PACE series polymers is that the density of cations is lower than that of most cationic polymers, resulting in lower toxicity; the existence of hydrophobic monomer PDL is helpful to effectively form particles at low cationic density [170].

Cui et al. proved that PACE nanoparticles (NPs) can provide a large amount of protein knockout in human embryonic kidney cells (HEK293) and primary human umbilical vein endothelial cells (HUVECs) that are difficult to transfect. Hydrophobic lactones with high monomer content are synthesized and accumulated in the liver, providing siRNA treatment opportunities for alcoholic liver disease and liver fibrosis [171].

By electrostatic interaction, cationic polymers combine with siRNA to form polymers with beneficial therapeutic effects in vivo and in vitro. Although the high charge of cationic polymers leads to the high loading rate of siRNA, its toxicity cannot be avoided. The functionalization of cationic polymers will overcome toxicity problems and remain the most important non-viral vectors in the future.

#### 2.2.3. Coordination Polymer

Coordination polymers (CPs) are composed of metal ions and organic ligands. As a promising vector for gene delivery, it has attracted great interest in the biological community. CPs have several advantages over conventional polymers, including adjustable size, clear pore size, high loading rate, and great biocompatibility [172]. In recent years, CPs have been used as vectors for gene therapy, carrying siRNA and DNA ribozymes, among others.

Liu et al. have prepared GSH-responsive photosensitive nanocarriers using [Cu(tz)] nanosheets to adsorb Ce6-DNAzyme on the surface of the nanosheets to form Ce6-DNAzyme/[Cu(tz)]. When endocytosed by cancer cells, Ce6-DNAzyme/[Cu(tz)] can be decomposed by GSH to release DNAzyme and catalyze the cleavage of the target mRNA. Ce6 produced ^1^O_2_ upon 660 nm laser irradiation (Figure 6). Furthermore, [Cu(tz)] nanoplates produced hydroxyl radicals under 808 nm laser excitation, and Ce6-DNAzyme/[Cu(tz)] demonstrated promising antitumor efficacy by using a combination of gene silencing, type II PDT and type I PDT [173]. Zhu et al. utilized two-dimensional (2D) coordination nanosheets loaded with oligodeoxynucleotides composed of Zn^2+^ and Tetrakis (4-carboxyphenyl) porphyrin (TCPP). Two-dimensional nanomaterials have a large surface area loaded with cytosine-phosphorothioate-guanine (CpG). SDT can work with Zn-TCPP/CpG to trigger the immune memory effect and inhibit tumor recurrence after the elimination of the primary tumor (Figure 7). Two-dimensional coordination nanosheets may be a promising and efficient platform for cancer treatment [174].

### 2.3. Inorganic Nanoparticles

Despite their high transfection efficiency and low production cost, Liposomes and polymers are limited by their toxicity, lack of targeting, and instability [175]. Inorganic nanoparticles have drawn much attention for drug delivery to specific sites since they have a high specific surface area and a surface that is easily chemically modified [176], providing a reliable platform for siRNA delivery. At present, the commonly used inorganic nanocarriers include gold nanoparticles (AuNPs) [177], graphene oxide (GO) [178], carbon nanotubes [179], and mesoporous silica [180].

#### 2.3.1. Gold Nanoparticles (AuNPs)

AuNPs are the most widely used siRNA delivery carriers in inorganic materials [181]. It has been proved that most AuNPs bind to siRNA through the Au-S bond. The advantages of gold nanoparticles as carriers include: AuNPs are non-toxic, biocompatible [182], and chemically stable in blood [183]; AuNPs have a high specific surface area and bind to siRNA in a particular manner [184]; AuNPs offer a possibility of the transfer and release of siRNA at specific sites by modification [185]. However, as carriers, a variety of factors, such as the size and shape of AuNPs, have an impact on the efficacy of transfection. In a study on the relationship between gold nanoparticle size and cell uptake, it was found that the 40–50 nm AuNPs had substantially higher cell uptake efficiency than 15 nm AuNPs [186]. Also, other studies have confirmed that spherical AuNPs have higher cell uptake efficiency than bar AuNPs [187].

AuNPs coated with oligonucleotides cannot distinguish tumor cells from normal cells, which hinders their application in treatment. Chang et al. formed an outer layer with tumor-targeting ability by laying adhesive Ap-YTDB on the surface of AuNPs loaded with siRNA. Sticky Ap-YTDB contains a y-shaped dsDNA backbone in the central, sticky terminal, and vertex-restricted aptamers, which can achieve the effect of release in target cells [188].

There are highly expressed glucose transporters1 (GLUT1) on the surface of malignant tumor cells. Yu et al. installed glucose on the α end of PEG-block poly (L-lysine) through an ether bond. The modification of glucose has the ability to recognize GLUT1 and endow Glu-NPs with the specific targeting ability to GLUT1 [189].

Inflammation is caused by macrophages secreting inflammatory factors that cause many autoimmune diseases. The inhibition of the expression of cytokines has proved to be beneficial to the treatment of inflammation. Jiang et al. designed a gold nanoparticle carrier to transport siRNA targeting TNF-α to the cytoplasm of RAW 264.7 cells. The results show that 80% or more NPSC/siRNA accumulates in the spleen, and the transfection rate in RAW 264.7 is as high as 80% [190].

Li et al. proposed a nuclear-targeted siRNA delivery system. The nano-carrier is composed of siRNA-modified AuNPs and nuclear localization signal (NLS) peptides. NLS peptide can interact with the nuclear pore complex and transfer the nano-carrier into the nucleus, thus achieving nuclear-targeted gene silencing [191].

#### 2.3.2. Graphene Oxide (GO)

In recent years, GO, as a nanocarrier, has attracted wide attention due to its high specific surface area [192], biocompatibility [193], and easy degradation in vivo without toxic substances [194]. The loose and porous structure of GO and its functional groups, such as hydroxyl, epoxide, and carbonyl groups on the surface of GO, enable it to bind to siRNA and chemotherapy drugs through π-π* stacking, electrostatic attraction, or other actions [195]. Additionally, studies have shown that in vivo administration did not cause various adverse symptoms [196]. The findings of these studies confirm that GO is an effective vector for gene silencing.

The hydrophilic groups on the surface of GO, such as hydroxyl and carboxyl groups, make GO readily functionalized by PEG with strong biocompatibility and improve the loading rate and biocompatibility of siRNA of GO. As well as being used as nanocarriers, GO’s photothermal effects are also widely utilized in cancer treatment [197]. Li et al. modified CRGDFV-functionalized GO with octagarginine (R8) in the cell-penetrating peptide (CPPS) to improve cell uptake efficiency and deliver VEGF-SiRNA into HeLa cells. In vivo and in vitro studies showed that the nanocarriers down-regulated VEGF gene expression and inhibited tumor growth [198].

Yang et al. constructed go-PEI-PEG-CPP nanocarriers, in which the addition of cell-penetrating peptide (CPP) improved cell uptake efficiency while PEI and PEG enhanced the binding ability to siRNA and Go-PEI-PEG-CPP-siRNA has demonstrated efficacy in the treatment of breast cancer at the cellular level and mouse models of breast cancer [199].

#### 2.3.3. Carbon Nanotubes (CNTs)

CNTs, as carriers of siRNA and drugs, have their unique advantages and have drawn much interest from the nanomedicine community [200]. First, CNTs have a high specific surface area to facilitate their interaction with ligands [201]. Secondly, CNTs have delocalized large π bonds on their surfaces that can be bonded to molecules with conjugated π bonds by π bonds. Finally, cell membranes can be effectively penetrated by CNTs. Multi-walled CNTs (MWNTs) were used as delivery carriers by Wen et al. to deliver SORafenib (Sor) and epidermal growth factor receptor (EGFR) siRNA into HepG2 cells. The results demonstrated enhanced uptake of MWNT/Sor/siRNA and increased Sor and siRNA release, proving that CNTs exhibit great anti-tumor effects [202].

However, CNTs have poor water dispersion and a high level of toxicity [177]. As a result, surface functionalization of covalent or non-covalent CNTs is crucial [203]. It has been demonstrated that cationic polymers can graft CNTs to improve their dispersity, reduce their toxicity, and improve gene transfection efficiency [204]. The most used polymer for CNTs modification is PEG. Hu et al. designed a cell-penetrating peptide and linked it to pegylated MWNTs to obtain MWCNT-PEG-TAT. They also coupled FITC to MWCNT-PEG-TAT to monitor the cellular uptake of the nanocervator. DOX is prepared for DOX-loaded stacking to obtain DOX-loaded nanocarriers. The data of particle size, polydispersity index, and zeta potential of MWCNTS-PEG, MWCNTS-PEG-TAT, DOX-MWCNTS-PEG, and DOX-MWCNTS-PEG-TAT are shown in Table 3. The results show that the nano-carrier has suitable particle size and stability. The drug loading rates of MWCNTs-PEG and MWCNTs-PEG-TAT for DOX were 97.3% and 98.2%, respectively. The nano-carrier showed a high drug loading rate for DOX. The pH-responsive release of DOX in PBS at 37 ℃ was investigated in vitro. The data shows that at pH 7.4, the release rate is close to 30% within 72 h, while under acidic conditions, the release rate is as high as 80% within 72 h. After co-incubation of 10 μg/mL DOX-MWCNTs-PEG-TAT with MG63 cells for 72 h, most cells experienced apoptosis, with 92.08% of cells observed in the Annexin V positive or PI positive quadrants [205].

#### 2.3.4. Mesoporous Silica (MSN)

Mesoporous silica nanoparticles (MSN) have attracted extensive attention in the biomedical field due to their large specific surface area, biocompatibility [206], and mesoporous-mediated loading capability [207]. One of the advantages of using MSN as a carrier is that nanoparticles can be passively concentrated at the tumor site by utilizing the EPR effect of the tumor [208]. The second advantage is that the MSN surface is easy to modify, and research has shown that surface modification can improve its therapeutic effect [209]. Mora-Raimundo et al. used PEI to function MSN to jointly deliver SOST siRNA and osteogenic statin to promote bone formation, which has great application prospects in the treatment of osteoporosis (Figure 8) [210]. To transport NH_2_-MSN without degradation, Heidari et al. coated siMDR1 with PEG and folate functionalized CS as a carrier (Figure 9). They then demonstrated that the cell uptake of nanoparticles in Hela-RDB cells with the folate receptor was twice as high as that of EPG85.257-RDB. In vivo results showed that the protein level of MDR1 was significantly decreased, and this nanoparticle offers excellent potential for use in the treatment of multidrug resistance in cancer [211].

As siRNA carriers, inorganic nanoparticles offer a practical application in gene therapy thanks to their obvious advantages, including the ability to protect siRNA, low immunogenicity, and easy surface modification. The modified surface of inorganic nanoparticles improves gene transfection efficiency as well as biocompatibility. Although inorganic nanoparticles have been used as gene therapy vectors only recently, they have received extensive attention in the field of gene therapy. As the research advances, inorganic nanoparticles will eventually become one of the ideal gene therapy vectors.

## 3. Conclusions and Prospect

As a new treatment method, gene therapy has far-reaching significance in the treatment of malignant tumors and congenital genetic diseases. Over the past decade, gene therapy has developed rapidly and made great progress. Even though most current methods of gene therapy employ viral vectors, there are still certain flaws, mutation risks, and limitations. New research on non-viral vectors has advanced tremendously and has been gradually applied in clinical practice. An excellent siRNA delivery system should first form a stable structure with siRNA that protects against nuclease degradation in the blood. Second, nanovectors can target cells and enter cells through endocytosis. Finally, nanovectors should be designed to successfully help siRNA escape from the endocytosome into the cytoplasm. Several commonly used non-viral vectors for siRNA delivery are described in this review, including various liposomes, biopolymers, synthetic polymers, AuNPs, MSN, CNTs, and GO and their functionalization. Cationic polymers are the most widely used among these carriers. Nanocarriers and their functionalization have created new avenues for the treatment of diseases. As a growing number of siRNA is expected to be developed, gene therapy will certainly advance tremendously soon.

Currently, malignant tumors are a serious threat to human health and are among the leading causes of human death. Tumors are mainly treated with chemotherapy, and most tumor chemotherapy drugs cannot eradicate the tumor. Even successful surgeries are plagued by recurrences. In most cases, chemotherapy fails due to multidrug resistance (MDR) in tumors. As an alternative, gene therapy has provided unimaginable results in curing tumors. The use of siRNA has a significant impact on the treatment of tumors by silencing the MDR-related gene expression and re-sensitizing the chemotherapy drugs. Through gene therapy, siRNA can be delivered to tumor cells together with chemotherapy drugs. Nanocarriers can protect siRNA from degradation and improve the efficiency of gene silencing. Nanocarriers can penetrate tumor tissue more easily through the EPR effect to achieve the targeted effect and can stay in tumor tissue for a long time. Drug delivery based on the EPR effect depends on various factors, including cycle time, targeting, and the ability to overcome obstacles, which depend on the size, shape, and surface characteristics of drug particles. The size of nanoparticles is a key factor. Research shows that the size range of about 40–400 nm has a longer cycle time and enhances the accumulation of tumors. Shape and form also play an important role in passive targeting. Generally speaking, rigid spherical particles with a size of 50–200 nm have a longer cycle tendency. Surface properties also play a key role in determining the internalization of drug particles into target cells. Also, fatal diseases such as pancreatic ductal carcinoma and hepatocellular carcinoma exhibited good therapeutic effects via gene therapy. This proves that gene therapy does open up new pathways of cancer immunity.

Most genetic diseases are difficult to cure and have a lifelong impact. Gene therapy has emerged as a new treatment option, with breakthroughs in basic research and clinical trials promising to make long-awaited gene therapy a reality for genetic diseases. Hemophilia is a condition in which blood fails to clot as a result of a deficiency in a certain clotting factor caused by bleeding or non-bleeding diseases. Hemophilia can be effectively treated using gene therapy. Adeno-associated virus (AAV) vectors target genes efficiently into specific cells or tissues, substituting therapeutic genes for pathogenic genes to cause minimal harm to the human body. It has been clinically effective in treating hemophilia using this approach. Gene therapy has also achieved success in the treatment of optic nerve genetic diseases, Pompey’s disease, Tay-Sachs disease, and sickle cell disease. Gene therapy has proven its effectiveness in genetic diseases.

Cardiovascular diseases (CVDs) are another main cause of death in the worldwide range. CVDs pose a severe threat to human life and health because of their high prevalence and mortality. Cardiovascular diseases mainly include coronary heart disease, angina pectoris, hypertension, and heart disease. Therefore, it is urgent to offer patients with cardiovascular diseases new treatment strategies. Gene therapy was investigated for this issue. Several clinical trials have demonstrated that gene therapy can be safely and effectively used for the treatment of heart dysfunction. Among many targets for heart failure treatment, AAV and retrovirus are the most commonly used vectors, which achieved excellent therapeutic effects on heart failure, coronary heart disease, and heart disease.

The breakout and rapid spread of COVID-19 since the end of 2019 can cause acute infectious pneumonia. Mild cases can cause fever and cough, while severe cases may result in respiratory failure, multiple organ damage, and even death. The ongoing spread of COVID-19 has posed new challenges to global health development, and biomedical science has become a priority area for countries.

The US FDA granted the authorization for mRNA-based Moderna company’s mRNA-1273 and Pfizer biotech’s BNT162b2 vaccine developed by lipid nanoparticles for the first time. Their successful implementation in clinical therapy demonstrated the promise of nanomedicine to address COVID-19 [212].

Gene therapy is a promising therapeutic strategy that overcomes the limitations of conventional therapies. The scope of gene therapy is already extensive, covering not only congenital genetic diseases, malignancies, and COVID-19 but also other conditions, such as cardiovascular and neurodegenerative diseases. The implementation of the human genome project will significantly expand the scope of gene therapy and bring profound influence on human health as new genes are discovered, and new technologies are developed. It is believed that the utilization of gene therapy in clinical trials offers exciting new prospects for the advancement and development of therapeutics.

## Figures and Tables

**Figure 1 ijms-24-03375-f001:**
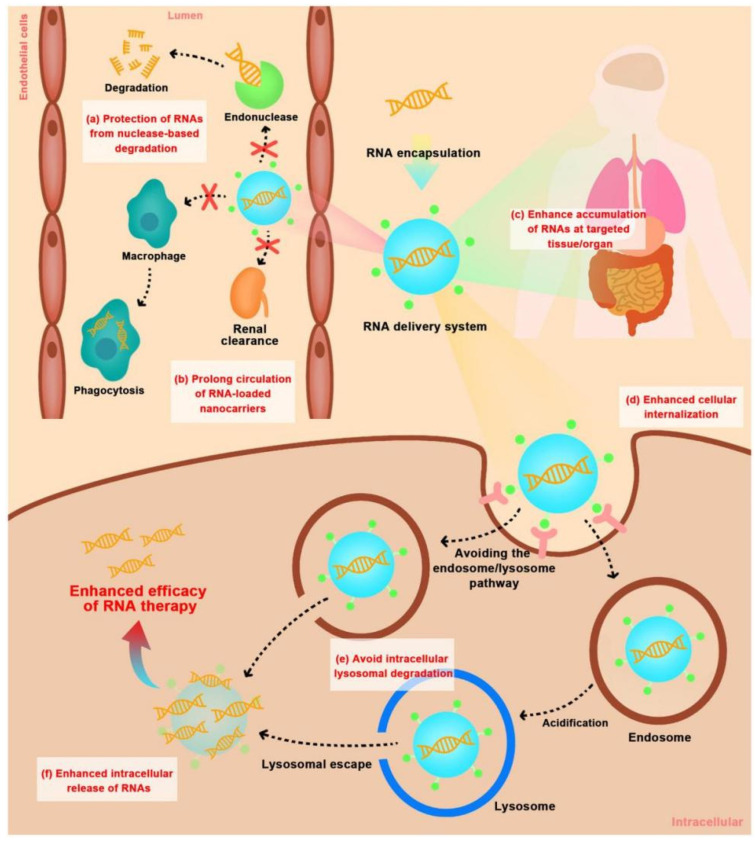
Extracellular and intracellular barriers for in vivo delivery of RNAs using non-viral vectors. (**a**) protection of RNAs from nuclease-based degradation; (**b**) prolong circulation of RNA-loaded nanocarriers by avoiding phagocytosis by the mononuclear phagocytic system and rapid kidney clearance; (**c**) enhance tissue/organ-selective accumulation of RNAs; (**d**) enhance cellular internalization; (**e**) avoid intracellular lysosomal degradation; (**f**) enhance the intracellular release of RNAs. Reprinted with permission from Ref. [34]. 2022, Elsevier.

**Figure 2 ijms-24-03375-f002:**
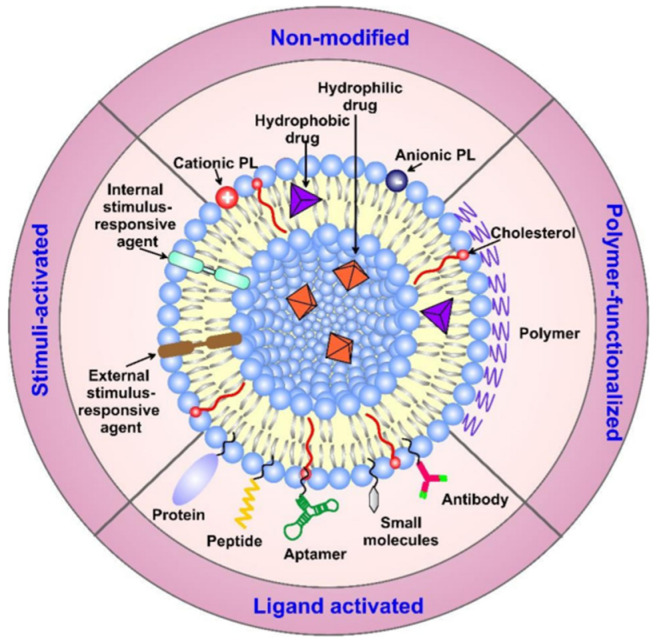
Surface modification strategies of liposomes, together with their classification. The modified carriers can contain active components, such as drugs, small molecules, proteins, and/or targeting moieties, such as antibodies, peptides, aptamers, etc., conjugated on the surface of the vehicles through different linkers, non-covalent or covalent bonds, and electrostatic interactions. Abbreviation: PL-phospholipid. Reprinted with permission from Ref. [87]. 2022, Molecular Diversity Preservation International.

**Figure 3 ijms-24-03375-f003:**
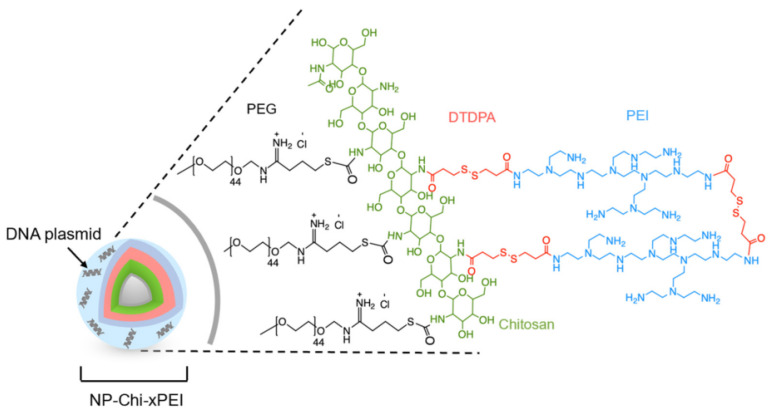
Schematic representation of NP−Chi−xPEI complexed with DNA with the zoom−in details of the structure of the Chi−xPEI polymer conjugated onto the surface of NP. Reprinted with permission from Ref. [120]. 2022, Molecular Diversity Preservation International.

**Figure 4 ijms-24-03375-f004:**
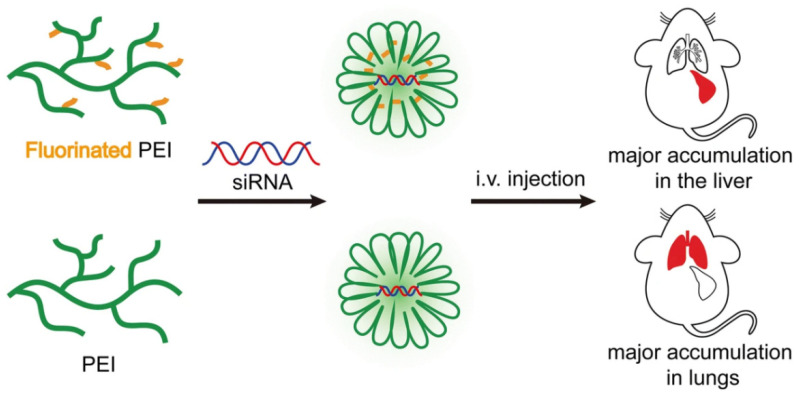
Fluorination of branched PEI (Mw = 25,000 g/mol) altered the biodistribution of siRNA from the lungs to the liver after intravenous administration of siRNA polyplexes to mice. Reprinted with permission from Ref. [143]. 2020, Springer Nature.

**Figure 5 ijms-24-03375-f005:**
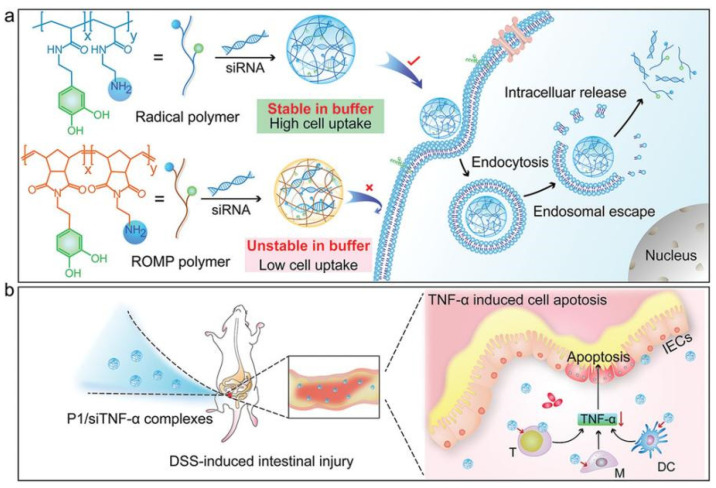
(**a**) Structures of the designed polymers and the proposed gene silencing mechanism in intracellular siRNA delivery. (**b**) A DSS-induced intestinal injury model was established to evaluate the gene knockdown efficiency of the lead radical polymer (P1)/siTNF-*α* complexes. The knockdown of TNF-*α* reduces DSS-induced intestinal epithelial damage. Reprinted with permission from Ref. [168]. 2021, John Wiley and Sons.

**Figure 6 ijms-24-03375-f006:**
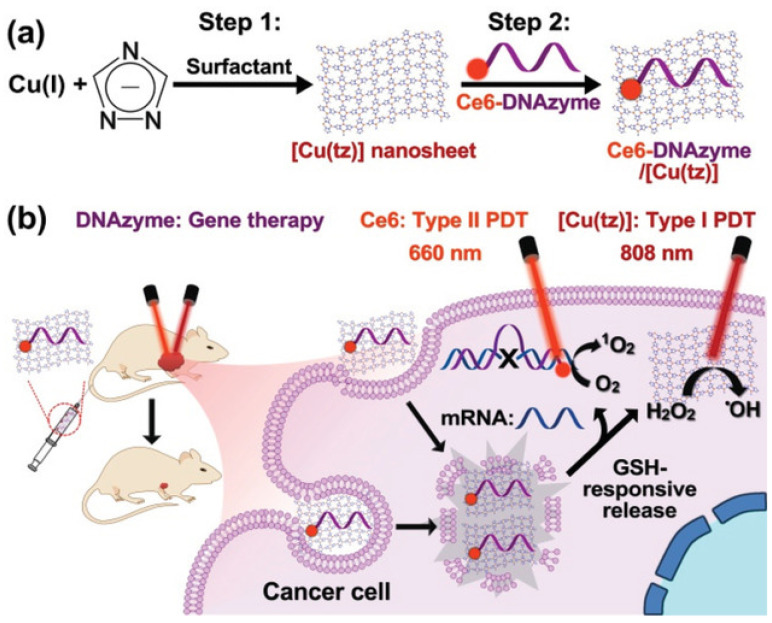
(**a**) Preparation of 2D [Cu(tz)] nanosheets and the Ce6-DNA-zyme/[Cu(tz)] therapeutic platform. (**b**) Schematic illustration of the proposed combination therapy of DNAzyme-based gene silencing, Ce6-based type II PDT, and [Cu(tz)]-nanosheet-based type I PDT. Reprinted with permission from Ref. [173]. 2021, John Wiley and Sons.

**Figure 7 ijms-24-03375-f007:**
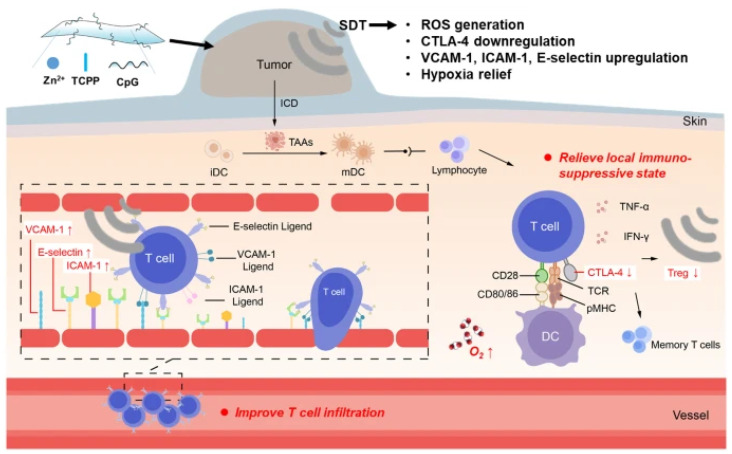
Schematic image of the synthesis of Zn-TCPP/CpG nanosheets and the mechanism of immune responses induced by Zn-TCPP/CpG-based SDT. Briefly, The SDT with Zn-TCPP/CpG could induce ICD of cancer cells and release TAAs, which could be engulfed and presented by DCs with the assistance of CpG. The immune system could be activated to suppress tumor metastasis owing to the antigen presentation by matured DCs. Moreover, ultrasound itself could strengthen anti-tumor immune responses by improving the tumor infiltration of T cells and limiting regulatory T cells in the tumor microenvironment. Reprinted with permission from Ref. [174]. 2020, Springer Nature.

**Figure 8 ijms-24-03375-f008:**
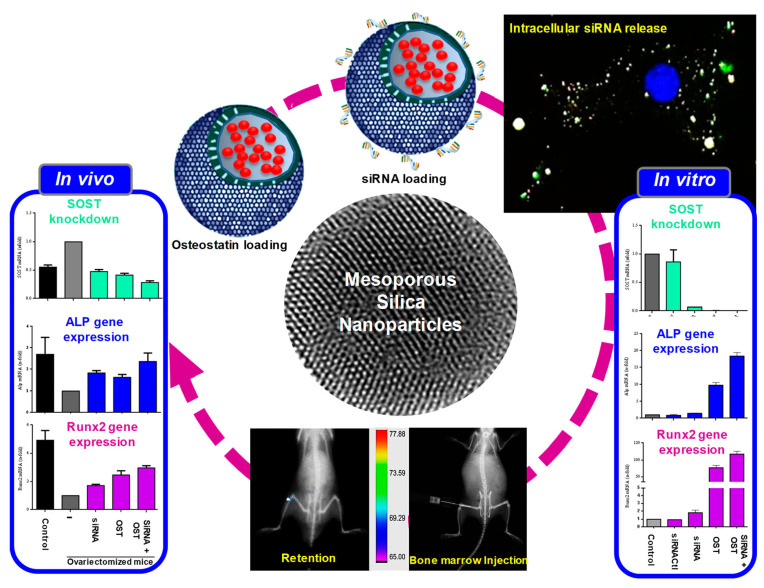
Schematic representation of the designed nanocarrier based on mesoporous silica nanoparticles loaded with osteostatin and siRNA to knockdown SOST and promote the expression of early markers of osteogenic differentiation both in vitro and in vivo. Reprinted with permission from Ref. [210]. 2019, American Chemical Society.

**Figure 9 ijms-24-03375-f009:**
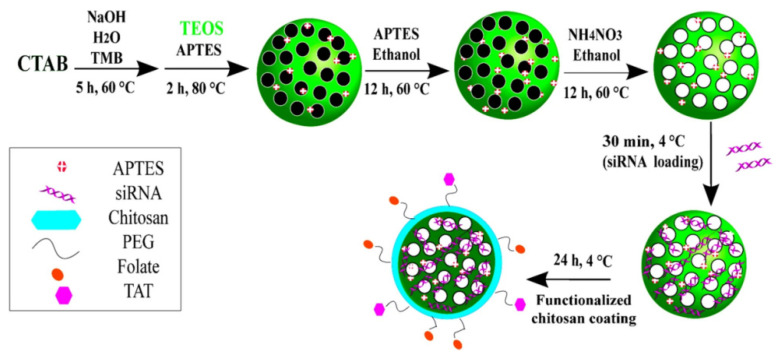
Schematic illustration of synthesis steps and mesoporous silica nanoparticles (MSNs) functionalization to obtain NH_2_-MSN, NH_2_-MSN-siRNA, and NH_2_-MSN-siRNA-chitosan functionalized with PEG-folate and PEG-TAT. APTES 3-Amino propyltriethoxysilane, CTAB cetyltrimethylammonium bromide, PEG polyethylene glycol, TAT trans-activator of transcription, TMB 1,3,5-triisopropylbenzene, TEOS tetraethyl orthosilicate. Reprinted with permission from Ref. [211]. 2019, American Chemical Society.

**Table 1 ijms-24-03375-t001:** A summary of the development of siRNA-based drugs with either FDA approval or in late phase 3 clinical trials. Reproduced with permission from [37].

Drug/Alternative Name	Company	Disease	Updated Status
Patisiran (ONPATTRO)	Alnylam	Hereditary transthyretin-mediated amyloidosis	FDA approval on 8 October 2018
Givosiran (GIVLAARI)	Alnylam	Acute hepatic porphyria	FDA approval on 20 November 2019
Lumasiran (ALN-GO1)	Alnylam	Primary hyperoxaluria type 1(PH1)	FDA approval on 23 November 2020
Vutrisiran (ALN-TTRsc02)	Alnylam	Hereditary transthyretin-mediated amyloidosis	Phase 3 trial
Nedosiran (DCR-PHXC)	Dicerna, Alnylam	Primary hyperoxaluria	Phase 3 trial
Inclisiran (ALN-PCSSC)	Alnyla, Novartis	Hypercholesterolemia	Phase 3 trial
Fitusiran (ALN-AT3sc ALN-APC SAR439774)	Alnylam, Sanofi Genzyme	Hemophilia A and B	Phase 3 trial
Teprasiran (AKli-5 DGFi I-5NP QP1-1002)	Quark, Novartis	Acute kidney injury Delayed graft function	Phase 3 trial
Cosdosiran (QP1-1007)	Quark	Non-arteritic anterior ischemic optic Neuropathy (NAION)	Phase 2/3 trial
Tivanisiran (SYL-1001)	Sylentis	Dry eyes Ocular pain	Phase 3 trial

**Table 2 ijms-24-03375-t002:** Properties of DOPC/DOPE, DOPC, and DOPC/Cholesterol liposomes. Reproduced with permission from [91].

Liposomes ^a^	Diameter (nm) ^b^	Zeta Potential (mV) ^b^	Lipid (mg/mL)	siRNA (μm)	siRNA/Lipids (%, *w*/*w*)
DOPC/DOPE	167	−14.1	0.23	3.24	16.2
DOPC	139	−6.7	0.56	3.91	8.5
DOPC/Cholesterol	100	−7.9	0.56	1.63	2.3

(^a^) Liposomes were suspended in PBS. (^b^) Diameter and zeta potentials were determined with Zetasizer Nano ZSP.

**Table 3 ijms-24-03375-t003:** Physiochemical characterization of different MWCNTs complexes. Reproduced with permission from [205].

Group	Particle Size (nm)	Polydispersity Index	Zeta Potential (mV)	DOX Loading Efficiency (%)
MWCNTs-PEG	89.39	0.343	−8.46	–
MWCNTs-PEG-TAT	96.26	0.394	−9.03	–
DOX-MWCNTs-PEG	135.24	0.290	−3.53	97.3
DOX-MWCNTs-PEG-TAT	145.24	0.339	−4.96	98.2

Statistical Analysis: The obtained results were expressed as mean ± standard deviation (SD) of triplicate.

## Data Availability

The data presented in this study are available on request from the corresponding author without any restrictions.

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
