# Peer review of "Research Status and Prospect of Non-Viral Vectors Based on siRNA: A Review"

_ijms, 2023, doi:10.3390/ijms24043375_

Round 1

Reviewer 1 Report

Authors proposed a paper entitled “Research Status and Prospect of Non-viral Vectors Based on siRNA: A Review” for the publication on the International Journal of Molecular Sciences, mdpi.

This paper is characterized by a quite good scientific soundness, and deserves to be published after major revisions.

The authors consulted a good amount of references in order to organize this review paper. I suggest adding some references about drug carriers classification and production methods, that could help expand the concepts expressed.

The introductive section ends with the definition of the scope of this paper, that could be expanded with a few lines more, in my opinion.

vaccine BNT162b2 shows 95% effectiveness in preventing severe COVID-19 this sentence should be supported by more references, since the value of 95% could be more or less effective depending on the period and covid variant. Please be more specific, adding information and references to this affirmation.

justice RNA please describe the meaning of this.

so the therapeutic effect; I would use therefore.

liposomes [28], inorganic nanoparticles [29] and polymers [30] are examples of some particles please check this expression. Here polymers are indicated as examples of particles such as liposomes and inorganic nanoparticles. I think this is not correct, since it would like like assessing that phospholipids are particles, instead of saying liposomes are particles. I suggest writing liposomes and inorganic and polymeric nanoparticles

Liposomes, also known as microlipids I think that this is not a particularly diffoused expression; therefore, I would suggest adding some references and explanations.

was liposome, thanks to their please check the concordance among liposome and their

Authors write about conventional liposomes, but they do not mention their preparation methods, that really are involved in their efficacy, bioavailability, solvent residue, mean size and so on. Please develop these concepts.

I suggest adding an abbreviation list due to the use of a large number of acronyms, according to the guidelines of this journal.

Escherichia coli this should be reported in italique and as E.coli.

Unmodified CS is the capital letter necessary with Unmodified?

TUMOR necrosis factor and achieved a positive why TUMOR is written in capital letters?

The experimental results illustrated that the concept nano carrier has the appropriate particle size, good stability, high drug load, good expression of anti-tumor effect I think that authors should be more specific in the definition of this information, giving data about mean size, entrapment efficiency and time stability.

NH2-MSN shoud the 2 be at th subscript?

In conclusion section, Gene therapy as a new therapeutic means will have far-reaching…” should be checked from the syntax point of view.

Author Response

  1. Response to comment: Adding some references about drug carrier’s classification and production methods, that could help expand the concepts expressed.Response: Thanks to the reviewers for their in-depth analysis and comments. However, this paper introduces the research status and prospect of gene therapy. The introduction explains the shortcomings of drug therapy, such as adding the drug carrier classification and production method will cause inconsistencies, and make readers think it is inconsistent with the title and other problems. Please see the yellow highlighted sentence in the revised manuscript on page 2.

  1. Response to comment: Expanding the conclusion of the introduction.

Response: In the introduction part, we add a summary of the previous introduction, and systematically discuss the main content and research prospect of this paper. Please see the yellow highlighted sentence in the revised manuscript on page 6.

  1. Response to comment: Adding the BNT162b2 vaccine’s information and references.

Response: We reviewed the literature and summarized the applicable population, dosage and effective time of BNT162b2 in this paper. Please see the yellow highlighted sentence in the revised manuscript on pages 2 and 3.

  1. Response to comment: Explaining what justice RNA means.

Response: In the second introductory paragraph, we introduce justice RNA. Please see the yellow highlighted sentence in the revised manuscript on page 2.

  1. Response to comment: Modified into liposome and inorganic polymer nanoparticles, the effect of liposome preparation method on particle size, encapsulation rate and bioavailability.

Response: In the section of traditional liposomes, we introduced the preparation method of liposomes and the effect of particle size on the targeting of liposomes, and detailed the information of preparation method, particle size, encapsulation rate and bioavailability of different literatures. Please see the yellow highlighted sentence in the revised manuscript on pages 7 and 8.

  1. Response to comment: Adding abbreviation table.

Response: Thanks to the reviewer's comments, we summarized the abbreviation at the end of the article. Please see the yellow highlighted sentence in the revised manuscript on pages 22 and 23.

  1. Response to comment: “Escherichia coli” should be reported in italique and as E.coli. “Unmodified CS” and “TUMOR” should be lower case. “NH2-MSN” should use the lower corner label.

Response: We have modified the parts involved. Thanks again for the reviewer's careful observation. Please see the yellow highlighted sentence in the revised manuscript on pages 9, 10, 14, 19.

  1. Response to comment: “The experimental results illustrated that the concept nano carrier has the appropriate particle size, good stability, high drug load, good expression of anti-tumor effect” I think that authors should be more specific in the definition of this information, giving data about mean size, entrapment efficiency and time stability.

Response: We consulted the literature to make a summary of the main content. Information such as particle size, potential and encapsulation rate are shown in the table. Please see the yellow highlighted sentence in the revised manuscript on page 19.

  1. Response to comment: Modified "Gene therapy as a new therapeutic means will have far-reaching...

Response: We have checked and adjusted this sentence. Thanks again for the reviewer's efforts. Please see the yellow highlighted sentence in the revised manuscript on page 20.

Reviewer 2 Report

Gene therapy is favored and valued by people because of its unique mechanism of action,

non-toxicity and good tolerance, which can kill cancer cells without damaging normal tissues.

Author Response

Response to comment: Some typos have occurred in the text. Please check if all abbreviations are included in the text. Perhaps some of them could be reduced.

Response: All spelling errors in the text have been corrected. All abbreviations in the text have been checked. And added an abbreviation list at the end of the paper. Please see the yellow highlighted sentence in the revised manuscript on pages 1-23.

Reviewer 3 Report

The authors in their paper titled:" Research Status and Prospect of Non-viral Vectors Based on siRNA: A Review" reported on the current status of siRNA application in treatments. Special attention is given to nanocarriers that help deliver siRNA into cells. The new results and implications are presented in this manuscript. Different formulations are suggested depending on the application needs. I have only minor comments. Some typos have occurred in the text. Please check if all abbreviations are included in the text. Perhaps some of them could be reduced.

Author Response

  1. Response to comment: In the abstract, the authors heavily focused on cancer therapy. It would be more readable if the authors mention the advantage of siRNA therapy on a wide range of diseases.

Response: The advantages of siRNA in the treatment of hemophilia, cardiovascular disease,

gastrointestinal disease and hepatitis B are added in the abstract. Please see the yellow highlighted sentence in the revised manuscript on page 1.

  1. Response to comment: The introduction started with mRNA vaccines while the rest part of the intro focused on cancer therapy. I suggest the authors shorten the first paragraph and merge the info with the fourth paragraph.

Response: The introduction of the first paragraph was modified, and the content of the first paragraph before the modification was simplified and combined with the fourth paragraph. Please see the yellow highlighted sentence in the revised manuscript on pages 2 and 3.

  1. Response to comment: Nanomaterials' accumulation through EPR effect faces challenges in human. Please discuss this in the conclusion part.

Response: In the conclusion part, we introduced the accumulation of nanocarriers in tumor tissue through the EPR effect and introduced the factors affecting the EPR effect. The influence of the size of the nanocarrier is introduced in detail. Please see the yellow highlighted sentence in the revised manuscript on page 21.

  1. Response to comment: Please provide detailed discussions about the translational potential for each delivery method.

Response: Thank the reviewers for their in-depth analysis and comments. The potential of each delivery carrier is discussed in each chapter. If you have any questions, please contact us.

  1. Response to comment: Please add a figure with the structures of liposomes.

Response: We added a diagram of the surface modification strategy and classification of liposomes, as shown in Figure 3. Please see the yellow highlighted sentence in the revised manuscript on page 8.

  1. Response to comment: Please include the following references.

Response: We have carefully read the following references and selected the parts suitable for the content of the article to quote. Please see the yellow highlighted sentence in the revised manuscript on pages 2, 17, 14, 15, 17.

  1. Response to comment: Several article, such as on page 9 second paragraph "...interaction, Cationic...". Please find and correct all of them.

Response: We have found and corrected the typos in this article. Please see the yellow highlighted sentence in the revised manuscript on page 15.

Reviewer 4 Report

Tong et al. systematically reviewed the recent approaches for siRNA delivery using nanomaterials. The authors stated the origin of siRNA-based therapy and provided their opinions on every delivery method. However, the following issues should be addressed before publishing:

1. In the abstract, the authors heavily focused on cancer therapy. It would be more readable if the authors mention the advantage of siRNA therapy on a wide range of diseases. 

2. The introduction started with mRNA vaccines while the rest part of the intro focused on cancer therapy. I suggest the authors shorten the first paragraph and merge the info with the fourth paragraph. 

3. Nanomaterials' accumulation through EPR effect faces challenges in human. Please discuss this in the conclusion part. 

4. Please provide detailed discussions about the translational potential for each delivery method.

5. Please add a figure with the structures of liposomes.

6. Please include the following references:

  (1) Nanomaterial overview: Matter, 2020, 3, 1948-1975; Adv. Drug Deliv. Rev., 2020, 156, 188-213.

  (2) Polymers: Nat Commun 2017, 8, 191; J. Control. Release 2019, 304, 259-267.

  (3) Gold nanoparticle section: Nat Commun,2021, 12, 2928; Chem. Sci., 2017, 8, 2816-2822; J. Control. Release 2019, 295, 268-277; J. Control. Release 2018, 283, 235-240.

7. Several article typos, such as on page 9 second paragraph "...interaction, Cationic...". Please find and correct all of them. 

Author Response

  1. Response to comment: Parts of RNA vaccines are poorly written.

Response: We reviewed the literature and summarized the applicable population, dose and effective time of BNT162b2. Please see the yellow highlighted sentence in the revised manuscript on pages 2-3.

  1. Response to comment: Language problems very huge.

Response: We have checked the language problems of the full text and revised it. Thank you for your careful reading. Please see the yellow highlighted sentence in the revised manuscript on pages 1-23.

  1. Response to comment: Not covered with as per the proposed title.

Response: We have revised some articles and added the corresponding contents of each title. Thank you for your careful reading. Please see the yellow highlighted sentence in the revised manuscript on pages 1-23.

  1. Response to comment: not covered why non-viral vectors are important compared to viral vectors.

Response: In the introduction, we introduce the disadvantages of viral vectors, such as high immunogenicity and high production cost. The advantages of non-viral vectors were further compared. Please see the yellow highlighted sentence in the revised manuscript on page 1.

  1. Response to comment: Tumor as main topic of siRNA development approaches. Not discussing any other disease or targets.

Response: In the abstract, we add the mechanism of siRNA therapy for hemophilia and neurodegenerative diseases. While presenting the various non-viral vectors separately, the literature is presented for the treatment of different types of cancer, chronic myeloid leukemia, Huntington's disease, etc. Clinical trials of siRNA have also included hepatic porphyria, acute kidney injury and dry eyes.  Please see the yellow highlighted sentence in the revised manuscript on pages 1, 22.

  1. Response to comment: Write the clinical trials or patents of non-viral siRNA case studies which were recently approved.

Response: We have summarized clinical trials or patents for non-viral siRNA case studies in Figure 1. Please see the yellow highlighted sentence in the revised manuscript on page 3.

Round 2

Reviewer 1 Report

Authors proposed another version of their paper, that has surely improved comparing to the previous one.

I suggest some minor revisions before publication.

Line 16. “normal tissues. It could be better to talk about non-carcinogenic tissues or healthy tissues?

Line 55. C. elegans this should be reported in italique.

Figure 1 should be named as Table 1. Please check this and correct the numeration of tables and figures. Moreover, check the guidelines for tables and figures of this journal; probably the headings of this table should not be colored. The focus of Figure/Table 1 is also poor: please report those information as table, not figure, loosing focus and quality.

Line 184. It can target maybe capital letter is not required here.

Line 187. And when the see previous comment.

Table 2. I would add standard deviations to define mean particle size more correctly.

Author Response

  1. Line 16. “normal tissues.” It could be better to talk about

non-carcinogenic tissues or healthy tissues?

Response: Thank you for your careful reading. We have changed "normal tissues" to "healthy tissues". Gene therapy has attracted much attention because of its unique mechanism of action, non-toxicity, and good tolerance, which can kill cancer cells without damaging healthy tissues.

  1. Line 55. “C. elegans” this should be reported in italique.

Response: Thank you for your careful reading. We have changed "C. elegans" to italic. In 1998, Mello et al. injected the mRNA-encoding muscle proteins into C. elegans, where the mRNA carrying the genetic code is called the "justice mRNA," and the mRNA carrying the antigenetic code information is called the "antisense mRNA."

  1. Figure 1 should be named as Table 1. Please check this and correct the numeration of tables and figures. Moreover, check the guidelines for tables and figures of this journal; probably the headings of this table should not be colored. The focus of Figure/Table 1 is also poor: please report that information as table, not figure, loosing focus and quality.

Response: Thank you for your careful reading. We have changed the name of Figure 1 to Table 1. And checked this and corrected the numeration of tables and figures. Table 1 A summary of development of siRNA-based drugs with either FDA approval or in late phase 3 clinical trials [37].

Drug/Alternative Name

Company

Disease

Updated status

Patisiran (ONPATTRO)

Alnylam

Hereditary transthyretinmediated amyloidosis

FDAapproval on 10/08/2018

Givosiran (GIVLAARI)

Alnylam

Acute hepatic porphyria

FDA approval on 11/20/2019

Lumasiran (ALN-GO1)

Alnylam

Primary hyperoxaluriatype 1(PH1)

FDA approval on 11/23/2020

Vutrisiran (ALN-TTRsc02)

Alnylam

Hereditary transthyretinmediated amyloidosis

Phase 3 trial

Nedosiran (DCR-PHXC)

Dicerna, Alnylam

Primary hyperoxaluria

Phase 3 trial

Inclisiran (ALN-PCSSC)

Alnyla, Novartis

Hypercholesterolemia

Phase 3 trial

Fitusiran (ALN-AT3sc

ALN-APC

SAR439774)

Alnylam, Sanofi Genzyme

Hemophilia A and B

Phase 3 trial

Teprasiran (AKli-5

DGFi

I-5NP

QP1-1002)

Quark,

Novartis

Acute kidney injuryDelayed graft function

Phase 3 trial

Cosdosiran (QP1-1007)

Quark

Non-arteritic anteriorischemic opticneuropathy(NAION)

Phase 2/3 trial

Tivanisiran (SYL-1001)

Sylentis

Dry eyes

Ocular pain

Phase 3 trial

  1. Line 184. “It can target” maybe capital letter is not require here.

Response: We have revised the problem of capital letters. We have revised the problem of capital letters. it can target liver tissue at 50-100 nm; 0.1-0.2 μm can target lysosomes of liver macrophages in liver tissue.

  1. “And when the” see previous comment.

Response: We have revised the problem of capital letters. We have revised the problem of capital letters, and when the particle size is greater than 15 μm, it can be absorbed by vascular epithelial cells such as mesenteric artery.

  1. Table 2. I would add standard deviations to define mean particle size more correctly.

Response: Thanks to the reviewer's comments. The results in the table have been expressed as mean ± standard deviation (SD).

Table 3. Physiochemical characterization of different MWCNTs complexes [205].

Group

Particle size (nm)

Polydispersity index

Zeta potential (mV)

DOX loading efficiency (%)

MWCNTs-PEG

89.39

0.343

−8.46

MWCNTs-PEG-TAT

96.26

0.394

−9.03

DOX-MWCNTs-PEG

135.24

0.290

−3.53

97.3

DOX-MWCNTs-PEG-TAT

145.24

0.339

−4.96

98.2

Statistical Analysis: The obtained results were expressed as mean ± standard deviation (SD) of triplicate.

Reviewer 2 Report

Manuscript modified as per comments advised.

Author Response

  1. Manuscript modified as per comments advised.

Response: We have revised every comment as required. Please see the yellow highlighted sentence in the revised manuscript on pages 1-19. Thank you for your careful reading.